# Stepped care, stepped care "lite" & matching intervention components to individual mental health needs: A rapid scoping review of mental health and substance use interventions for post-secondary students

**Sarah Brennenstuhl**[1,2], **Celeste Agard**[2], **Rachel Ho**[2], **Kristin Cleverley**[2,3,4]

**1** Student Mental Health, University of Toronto, Toronto, Canada, **2** Inlight, Student Mental Health Research Initiative, University of Toronto, Toronto, Canada, **3** Lawrence S. Bloomberg Faculty of Nursing, University of Toronto, Toronto, Canada, **4** Centre for Addiction and Mental Health, Toronto, Canada

* Sarah.brennenstuhl@utoronto.ca

## Abstract

### Purpose

Stepped Care Models (SCM) and other approaches for organizing the delivery of services and resources by individual mental health (MH) needs are being increasingly implemented in post-secondary institutions. However, no consensus definitions exist of what constitutes a SCM for post-secondary students (PSS), and there is little guidance for evaluation of these complex, multicomponent interventions. The purpose of this research is to identify and characterize MH and substance use interventions for PSS that apply a SCM, stepped approach (i.e., stepped care "lite"), and/or organize delivery of resources/services based on individual MH needs.

### Methods

A rapid scoping review of peer-reviewed research articles was conducted using OVID MEDLINE®, OVID Embase, EBSCO CINAHL, OVID PsycINFO®, and ERIC. Eligible studies included multicomponent interventions for improving MH or substance use among PSS applying a SCM, stepped approach or another way of organizing resources/services offered according to individual MH needs. Results: 5757 abstracts were reviewed, resulting in full text examination of 172 studies. Data were extracted from 68 eligible studies comprising 50 interventions (SCMs: n=7, stepped care "lite": n=13; organized delivery matched to MH needs: n=30). Almost all involved a website/app and symptom tracking was often included within the intervention. Most addressed either alcohol use, depression, anxiety and eating disorders. A variety of evaluation models were applied, but approaches were not generally geared to look at individual-level outcomes in a manner that captured the overall effect of the SCM or outcomes related to the specific "dose" of the intervention received. Most outcomes focused on MH symptoms, satisfaction, and utilization; student-related outcomes such as academic success were rarely used. Student co-design was not often described.

**Data availability statement:** All relevant data are within the paper and its Supporting information files.

**Funding:** The author(s) received no specific funding for this work.

**Competing interests:** No authors have competing interests.

## Conclusions/Implications

Despite increasing implementation of SCMs in post-secondary settings, few studies on the model have been published. Drawing on strengths and shortcomings of studies identified, recommendations for future work in this area are presented.

## Introduction

With the rising mental health needs of post-secondary students (PSS) [1–4], exacerbated by the COVID-19 pandemic [5] and limited resources in post-secondary settings compared to demand [4,6], a reimagining of campus-based mental health services may be required [7]. A stepped care approach has been recommended for guiding redesign efforts across post-secondary institutions (PSI) [7,8]. It is argued that Stepped Care Models (SCM) for PSS should be formulated on the basis of matching the appropriate type and intensity of care with an individual's needs, preferences and readiness to change, and then stepping up or down treatment intensity as these vary [7–9]. This can be achieved through provision of timely and open access to a variety of services and care options designed to meet varying levels of functioning, commitment, and willingness to engage [10]. Assuming most students will need to access only lower intensity resources and services, the most resource-intense options (e.g., speciality treatment) can be preserved for the minority of students with the most complex needs. This provides an efficient method for service delivery that maximizes limited resources while meeting the diverse needs of students [7,8]. In addition to enhancing efficiency, making services and supports open access creates an opportunity for students to exercise agency in accessing care, as well as addressing health care needs at a population-level by providing upstream mental health prevention and promotion resources [10].

SCMs for delivery of mental health services have been implemented to varying degrees across a variety of PSIs [8,11,12]. While the literature on SCMs for mental health care delivery among youth has recently been reviewed [9], as far as we know there has yet to be a comprehensive attempt to survey the literature relating to PSS specifically. This knowledge gap is concerning as increasing implementation of SCMs on post-secondary campuses is expected, given budgetary constraints in the sector [13] and the rise of consulting firms such as Stepped Care Solutions [14] that help PSIs implement SCMs. Systematic gathering of evidence on efficacy of these models is needed, but before that can happen, an understanding of the landscape of SCMs for delivery of mental health interventions in the post-secondary sector is required.

Defining SCMs for the post-secondary sector is a challenge. In a descriptive overview of mental health services provided across Canadian PSIs, it was observed that application of SCMs was not fully explained by those reporting their use [11]. Further, without clear reporting on SCMs in PSIs, it is hard to develop a definition specific to this sector. Similarly, with reference to the broader literature on SCMs for youth, it was concluded that there is no consensus definition of SCMs, citing evidence of its broad application [9,15]. In a scoping review of SCMs for mental health care delivery among youth conducted by Berger et al., for the purposes of the review, SCMs were defined as: "systems of care" that must contain at least one "psychological treatment" of varying intensity and/or offered in more than one modality [9]. In addition, care recipients had to be "systematically evaluated according to defined improvement criteria" and care had to be adjusted if symptoms did not improve after an initial treatment [9].

In the post-secondary sector, the above definition may be too clinical to capture the types of interventions that are key to improving efficacy in service delivery while meeting the broad needs of students based in the community. For example, based on the literature, there appears

to exist a category of interventions what might be termed as stepped care "lite" approaches. While not meeting the definition of Berger et al. for SCMs [9], these approaches offer similar benefits with respect to efficiency and individualized care. They do this by applying an approach that can "flag" students who are using a lower-intensity intervention (e.g., an app) who experience a decline in symptoms and require higher-intensity care (e.g., individualized counselling) [16–18], or triage the intensity of the intervention provided according to symptom assessment [19–21]. For this type of stepped approach, clinical decision making tends to be algorithmic, achieved using technology that can track symptom levels and declines over time (i.e., apps), and triage care accordingly. Symptom tracking methods are among the lower intensity interventions recommended in some SCMs for PSS [8], and have become increasingly prevalent in mental health apps for young people [22].

Another application of symptom tracking is the ability to use the data to tailor the components of the intervention to the unique mental health needs of the individual. This ability to organize the delivery of services/resources by individual mental health needs is consistent with a key aspect of SCMs as they have been proposed for PSS [8]. While not necessarily varying the intensity of care, these approaches offer enhanced efficiency and individualized care, and thus may expand on or complement SCMs.

In addition to better understanding the landscape of SCMs and related approaches for delivering mental health interventions to PSS, guidance on how to evaluate these complex interventions is needed. While work of this nature has begun focusing on SCMs for youth more broadly [15], there needs to be clear recommendations for the post-secondary environment specifically. Compared with dedicated mental health systems, PSIs are more constrained in their ability to deliver mental health services, as evidenced by different funding models and the lack of specialized services, such as psychiatrists, on campuses [11]. Yet, PSIs have a very strong mandate from students to meet the needs of the entire population [23], including those who with highly acute clinical needs, sub-clinical needs (e.g., those experiencing academic-related stress) and those who may never engage with broader systems (e.g., international students). Evaluation models therefore need to account for these complex system- and population-based factors. This relates to the recommendation that SCMs be evaluated as a whole and not independently at each step [15,24]. Shifting the perspective to the overall effect of the model means evaluation designs must account for the potential for everyone to receive a different dose of the intervention comprising the unique combination of steps (or intervention components) received. Dosing may reflect individual mental health needs, but also levels of readiness and ability to engage in the intervention. It is not clear what evaluation methods are being used to assess SCMs and related approaches for PSS.

Continuous co-design with PSS and other key knowledge users such as service providers is thought to be essential to effective implementation of SCMs for PSS [10]. Co-design draws on lived experience to identify opportunities and develop strategies for intervention refinement, as well as to ensure user satisfaction and achievement of desired outcomes [25,26]. Experience-based co-design may also best enable the integration of the principal of cultural competence into mental health service delivery for PSS [7]. However, it is not clear to what extent those implementing SCMs and other stepped approaches among PSS are integrating student co-design. Therefore, in developing an understanding of the landscape, attention must be paid to whether, and to what degree, students are involved in the research process from conceptualization to dissemination.

Before any systematic attempts are made to assess the efficacy of SCMs for delivering mental health interventions to PSS, a consensus definition of SCMs must be developed [15] that can be applied to the post-secondary sector specifically. To do this, a better understanding of the landscape of SCMs, and related approaches being used is required. Therefore, the

objectives of this rapid scoping review of the peer-reviewed literature are two-fold: 1) to identify what mental health interventions for PSS exist that apply an SCM, use a stepped approach, and/or organize delivery of resources/services based on individual mental health needs; and, 2) synthesize the characteristics of the interventions, with a focus on evaluation methods used and student co-design.

## Materials and methods

### Eligibility criteria

**Study design.**  Rapid reviews are a simplified systematic review process that is used to facilitate a more timely knowledge synthesis [27]. A rapid approach is deemed suitable in instances where timely access to health information is required to inform decision making on pressing matters [27,28]. Given that SCMs have already been implemented in numerous PSIs across North America [8,11,12], use of a rapid review process to synthesize existing knowledge will provide timely guidance. To facilitate the rapid approach, the sole focus was peer-reviewed literature. Specifically, research articles and protocols reporting on multicomponent, mental health or substance use interventions for PSS based on a SCM, stepped approach, or organization of delivery of resources/services based on individual mental health needs were included. Definitions of these terms are provided in Table 1, and were developed using the literature in consultation with team members, including two mental health researchers, a clinician-scientist and an undergraduate student. To incorporate a student-perspective, the student also played a large role in the data extraction and interpretation.

Interventions featuring individual counselling alone, delivery of intervention components based on preference alone or individual characteristics unrelated to mental health needs (e.g., character traits [30]) were excluded, as were interventions for which it was unclear if the tailoring was based on individual mental health needs. One intervention was excluded because it was not clear whether the escalation process for addressing higher risk was part of the intervention design or only included for mitigation as part of the trial [31]. To facilitate a more rapid review, Personalized Normative Feedback interventions were excluded when that was the only method of delivering intervention components as this literature has already been extensively reviewed [32–34].

**Participants.**  Research articles reporting on PSS populations were included. PSS were defined as individuals enrolled in a 2- or 4-year first-degree program, graduate-level program, or professional program, and may be also referred to as college or university students. Those involved in specialized training outside of degree programs, such as a medical residency programs, were excluded.

**Language.**  Only articles reported in the English language were included.

### Information sources

**Electronic searches.**  The search was conducted on April 18, 2023, with an update on June 13, 2024, in OVID MEDLINE®, OVID Embase, EBSCO CINAHL, OVID PsycINFO®, and ERIC. Electronic search strategies were developed in consultation with a health sciences librarian. The search was not restricted for time period or geographic location. The full search strategy used for OVID PsycINFO® is available in the Supplemental Materials.

### Study selection process

The results of the search were uploaded to Covidence, an online program that aids in data extraction and management, and collaboration among team members [35]. Using a pair of independent reviewers (SB & CA), record titles and abstracts were screened for inclusion;

**Table 1. Definitions Used to Guide Scoping Review.**

| Term | Definition |
| --- | --- |
| *Mental Health Intervention* | A stand-alone intervention focusing on improving mental health symptoms (e.g., depression, anxiety, post-traumatic stress, psychological distress, self-harm and suicidal behaviours or symptoms related to an eating disorder, or substance use). Evaluation of the intervention must include at least one mental health or substance use-related measure as a primary outcome. As stress is not generally considered a mental health problem in and of itself, interventions focused solely or primarily on stress to the exclusion other mental health symptoms were not included in the review. |
| *Multicomponent intervention* | An intervention with multiple components, each theorized to have a distinct effect on an outcome [29]. A component can be as simple as a specific health message or as complex as entire treatment type. Consistent with Cornish's description of SCMs for PSS as organizing delivery of services/resources according to individual needs [8], each individual must have the potential to be exposed a different number and/or type of components within the multicomponent intervention. |
| *SCM* | Self-identified by the intervention authors that they applied a SCM model. This definition was selected in response to the observed heterogeneity in the design of SCM models in the literature [15,24], and our desire to be inclusive of all models that might fit in this category without being limiting. |
| *Stepped care "lite"* | A type of multicomponent intervention (see above definition) that provides two or more components based on mental health needs, with at least one component being higher intensity than another, without explicitly stating a SCM was applied. This could include a multicomponent intervention with: a) different models of care (e.g., a self-directed app that steps up to a phone call with a care provider); b) different modalities of the same care model (e.g., group counselling stepping up to individual counselling); or c) different types of information (e.g., information provided to everyone that steps up to an invitation for specific individuals to use services). This definition draws on a core component of Berger's definitions of SCMs of providing interventions at different levels of intensity [9] and Cornish's description of SCMs for PSS that includes organizing delivery of services/resources according to the mental health needs of the individual alone or in combination with preferences and level of readiness [8]. |
| *Delivery of resources/ services by individual mental health needs* | Consistent with Cornish's description of SCMs for PSS [8], this includes any individual-level mechanism for matching intervention components offered with the mental health needs of the individual, including provision of tailored feedback. Personalized interactions such as use of individual "e-coaches" or "supporters" fits this definition when the interactions have the potential to steer the individual to intervention components suited to their mental health needs but not if the sole purpose is to increase adherence or motivation. |

conflicts were resolved through consensus. Full-text reports were reviewed to confirm inclusion based on information from the title and abstract, and to screen records with an available abstract. References were found within the full-text reports. When multiple papers were available on the same intervention, the most definitive study reporting on intervention effectiveness was used for extraction, although any background information on the intervention itself was assessed if pertinent. If a definitive study was not available, the pilot study or protocol was used.

## Data collection and management

Forms to facilitate data extraction within Covidence were developed by the team and pilot-tested before real data was captured. Using a pair of independent reviewers (CA & RH), data was extracted and verified by a third reviewer (SB). Data extracted included study characteristics (e.g., date, country), population information (e.g., inclusion criteria), intervention details (e.g., mental health issue addressed, modality, mechanism for matching to mental health needs), evaluation framework used (e.g., study design, outcome measures, sample size), and student involvement.

### Data synthesis

The study characteristics were summarized using tables and described narratively.

## Results and discussion

In the original review and the update, a total of 5757 unique abstracts were identified using our search terms. Of these, 172 were screened into full text based on the title and abstract. Most studies excluded were done so because, in addition to not applying a SCM or stepped approach, they did not feature organization of the delivery of intervention components by individual mental health needs according to our definition. Data were extracted from 68 eligible studies comprising 50 interventions. See Fig 1 for the PRISMA flowchart.

Of the 50 interventions selected, a total of seven were stated to be based on a SCM, 13 studies were categorized as stepped care "lite", and 30 did not have a stepped approach but included a mechanism for organizing the delivery of intervention components based on individual mental health needs. The interventions were designed and/or undertaken across 13 different countries, the overwhelming majority coming from the United States (n = 28). Most interventions focused on general populations of students with or without specific clinical features (e.g., meeting a certain symptom level); five interventions, all based on the *StudentBodies* program, included women only [36–40]; two interventions focused on demographically unique populations of students (i.e., Chinese-speaking international students [41] and Arab students in Israel [42]). Publication dates ranged more than 20 years from 2000 to 2024, but half of the interventions were published in the last five years (25/50).

Most interventions across the three groups addressed alcohol use, followed by anxiety, depression and eating disorders. Three interventions addressed multiple mental health issues associated with student populations without specifying the conditions specifically [8,12,41]. Smoking and distress were the only issues addressed solely within the third category of studies focusing on delivery of intervention components based on mental health needs. Type of mental health issue addressed by intervention type can be found in Fig 2. Eight interventions explicitly stated that they were trans-diagnostic, meaning they addressed common elements of several underlying conditions [16,17,43–48].

Almost all interventions involved a website or app; most SCM and stepped care "lite" interventions involved counselling, whether in person or virtually, as higher intensity treatment options. The least often used modalities included Zoom videoconferencing [17,49] and workshops [8]. Intervention modality by intervention type can be found in Fig 3.

### Intervention details

*SCMs:* The 7 SCMs identified ranged from two to nine steps [8,12,50–54]. Five interventions had a specific algorithm for stepping up based on symptom measures [50–54], one intervention stated that the decisions about steps are made as a collaborative process between the counsellor and student [8], and one intervention indicated that treatment planning occurred during an initial consultation [12].

*Adapted Stepped Care* and *Stepped Care 2.0* are integrated SCMs involving an entire system of campus-based mental health services addressing a range of issues [8,12]. The interventions included eight [12] and nine [8] steps ranging from low-intensity interventions such as online information to outpatient therapy in *Adapted Stepped Care,* and psychiatric consultation and case management in *Stepped Care 2.0.* Neither intervention appeared to have an objective algorithm for stepping up but rather decisions were made on an individual basis with the student in consultation with a counsellor.

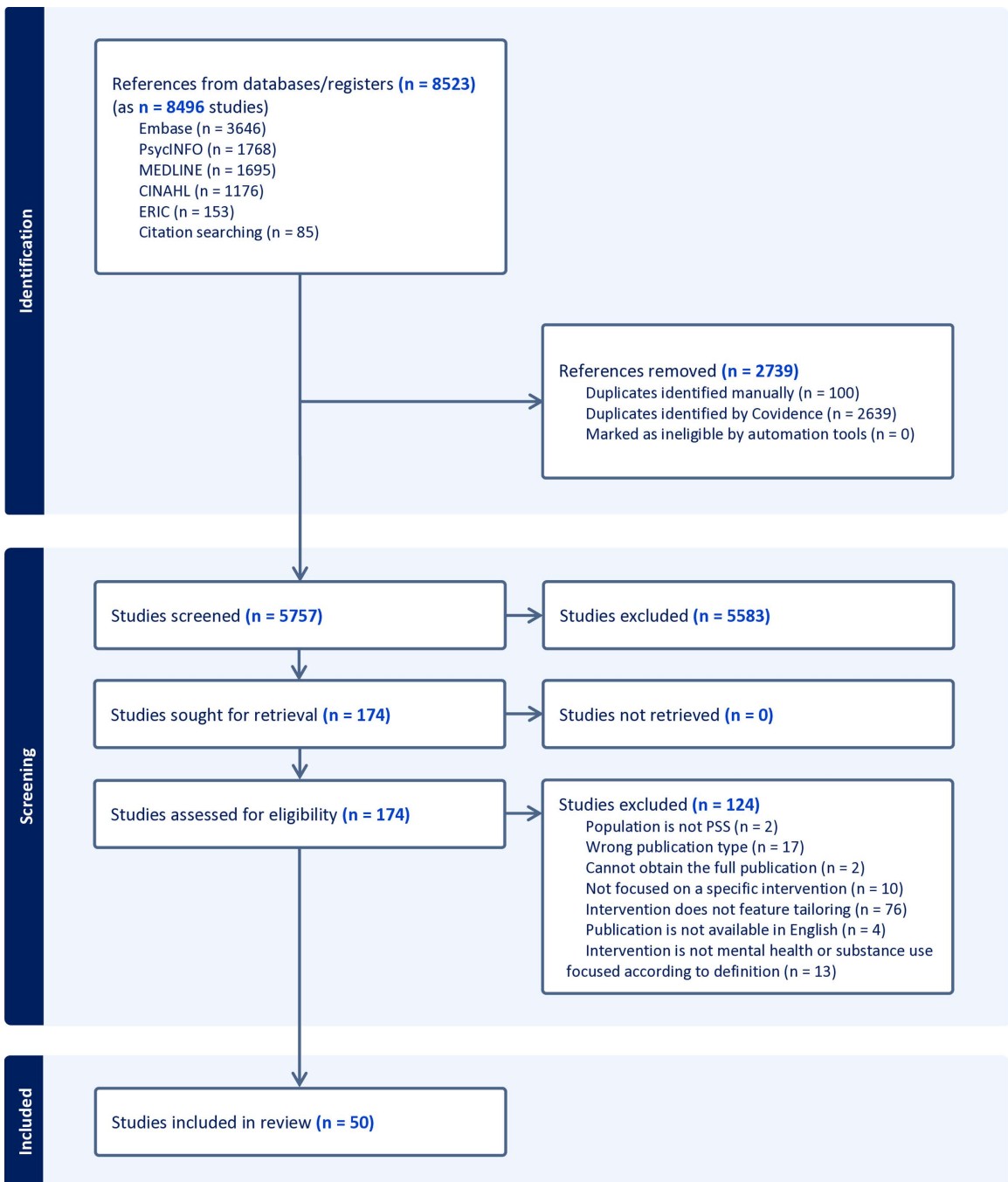

**Fig 1. PRISMA flowchart.**

*Screening and Treatment for Anxiety and Depression (STAND)* is a stratified SCM with four tiers of care that can be adapted based on continual monitoring of symptoms [54]. Tier 0 includes monitoring for those in the normal range of symptoms, tiers 1 and 2 offer online therapy with certified support for those with mild to moderate symptoms, and tier 3 comprises clinical care from psychologists and psychiatrists for those with severe symptoms.

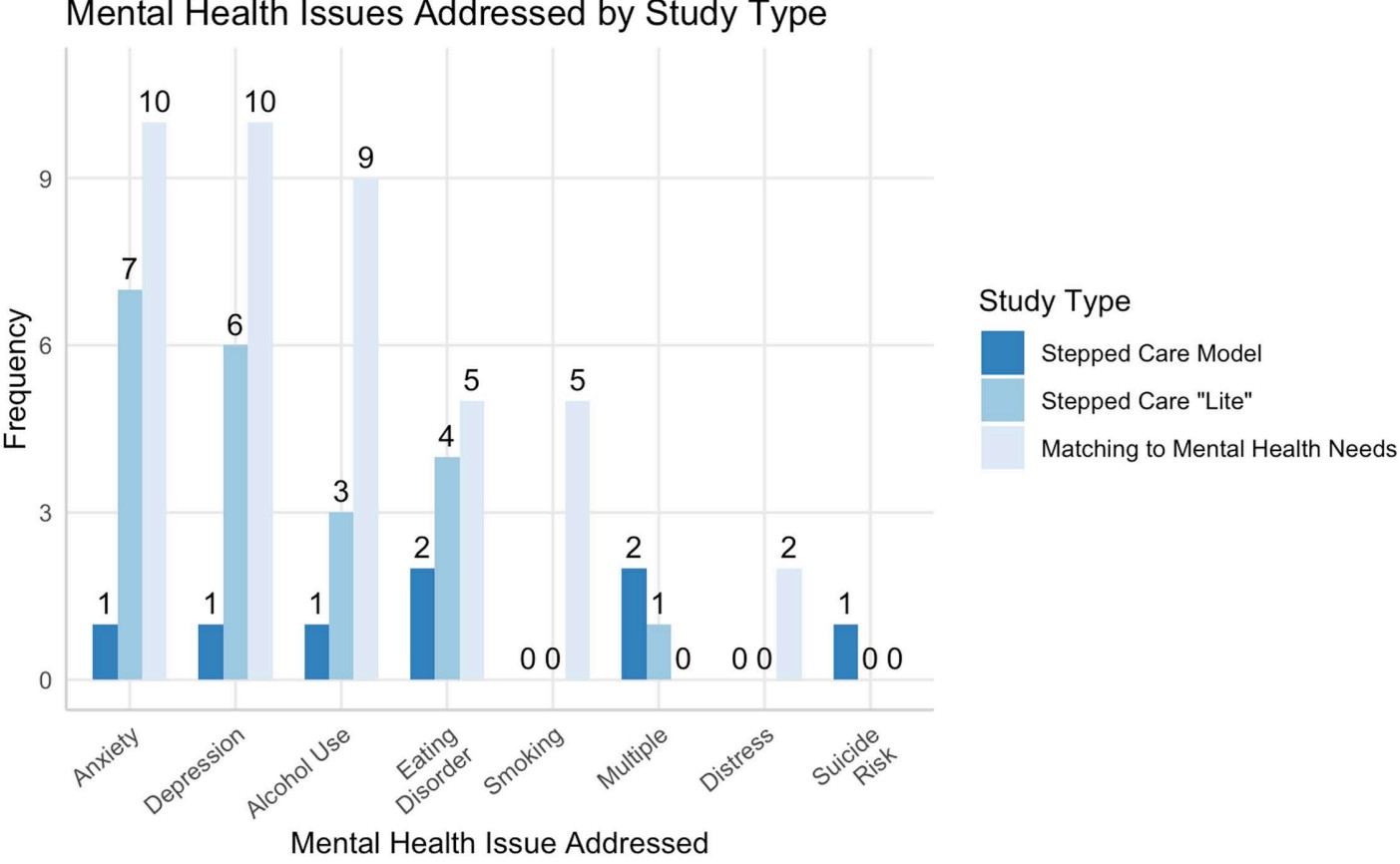

**Fig 2. Interventions by mental health issues addressed.**

*COMPAS-S* is an algorithmic intervention integrated into a local administration of the World Mental Health Surveys International College Student Initiative (WMH-ICS) to identity students at risk for suicidal behaviour [53]. Those reporting 12-month suicidal ideation are provided access to a safety planning app and those identified as high risk by a predictive algorithm or by reporting 12-month suicide attempt are contacted by telephone within 24 hours of completing the survey.

Two interventions are based on the ES[S]PRIT SCM model for sub-clinical eating disorders [50,52]. This model was implemented as part of the suite of campus-based mental health services and contained four steps: screening, online program participation, virtual therapist contact, and in-person counselling referral [50,52]. Decisions to step up are algorithmic, based on objective clinical indicators collected through weekly symptom tracking (e.g., body mass index, bingeing).

The last SCM intervention was designed for PSS mandated to an intervention due to violating campus alcohol policies. This intervention contained two steps: manualized brief advice with a peer counsellor for all participants, and a manualized Brief Motivational Interview provided by a professional counsellor with a personalized report for high risk drinkers [51]. After 6 weeks, PSS assessed to be at a high risk were assigned to the higher step based on an objective decision rule informed by data on drinking episodes and alcohol-related problems.

*Stepped Care "Lite:"* The 13 interventions in this category are characterized by their ability to flag PSS who require higher intensity interventions and generally contained two "steps." In 11 out of 13 interventions, this ability to "flag" higher intensity needs involved some type of

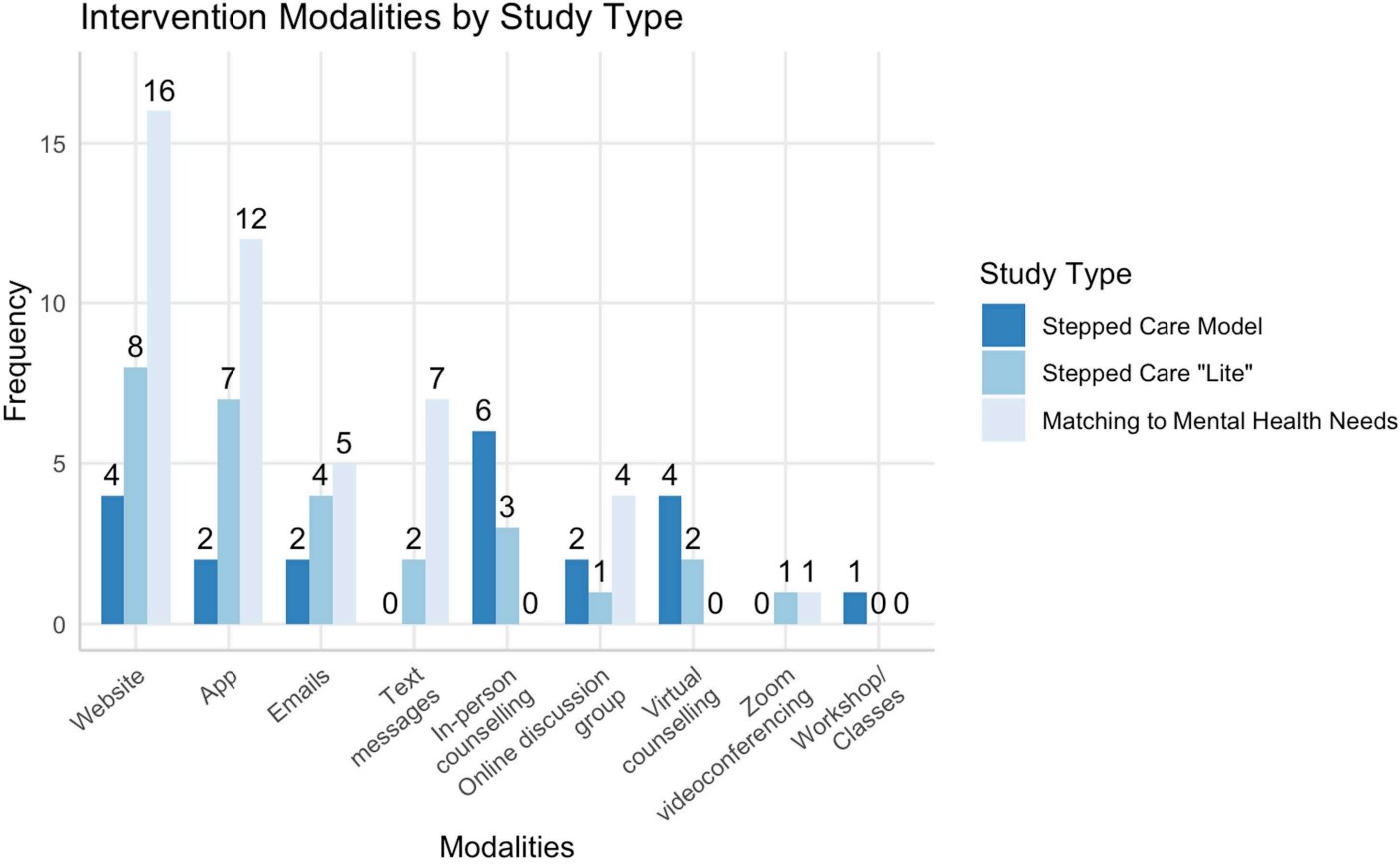

**Fig 3. Interventions by modality.**

ongoing symptom tracking. For example, in three interventions, therapist contact was initiated or stepped up [17,18,55] when symptoms were detected to decline. In eight interventions, a personalized list of campus-based resources [21,41,56] or referral to university counselling services was provided [16,36,57–59] based on symptom profile or those scoring in high-risk ranges. In a more unique style, one intervention focusing on alcohol use included two "stages." Stage 1 contained personalized normative feedback offered to all students. Stage 2 provided those identified as high risk based on bi-weekly self-monitoring surveys with either a resource email, or an invitation to chat with a health coach [19]. Finally, one intervention described as an "alternative" to a SCM, altered the modality of a Brief Motivational Interview for first time violators of campus alcohol policies requiring higher-intensity care. In one step, a Brief Motivational Interview was provided in a group format and in step two, an individual session was offered for those identified as high risk [20].

*Matching to Mental Health Needs.* The 30 interventions in this group were characterized by containing a mechanism that could match intervention components offered with the mental health needs of the individual PSS. In 19 interventions, the matching process was facilitated using some type of symptom tracking. The most common intervention design in this category (n = 14) involved providing tailored feedback and information based on assessed symptom profiles [37–39,42,43,46,47,49,60–65]. This was achieved most often within the content of the website itself, but in the *StudentBodies* interventions, the virtual feedback was provided by a human group moderator [37–39] and in *iCare Prevent*, *Best-U* and *Space from Depression*

*and Anxiety* interventions, by a virtual human "e-coach." [46–48,66] Also common within this group were interventions that provided activities, activity suggestions or specific web content that matched assessed symptom profiles (n = 12) [40,44–49,61,67–70]. Again, usually this was done within the content of the website, but in three interventions it was achieved via "e-coaches" or "supporters." [40,69,70] To improve therapy progression, an app called *elona therapy* was used alongside individual cognitive behavioural sessions; the content of the app was tailored by the therapist according to the student's diagnosis and individual mental health needs [71]. A sub-group of interventions within this category used text messages to achieve matching based on mental health needs (n = 5); three used text messages to address indicated or predicted urges to smoke [72–74] and two used text messages to address actual drinking behaviour and consequences [75,76].

## Symptom tracking

In total, 34 interventions incorporated some type of symptom tracking. However, how it was incorporated varied greatly with respect to timing of assessment. A total of nine interventions used daily or more frequent tracking [18,36,44,45,62,63,66,72,74], of which four explicitly mentioned Ecological Momentary Assessment (EMA) [44,45,62,72]. Another eight interventions used a weekly or biweekly schedule [16,17,19,38,50,52,54,55], while three used tracking at each session [57,58,77]. Finally, 13 interventions offered tracking that appeared to be optional and/or could be used at any time desired by the participant. All but two interventions [17,55] tracked using technology, using either in-app or web-based assessments.

## Evaluation methods

See Table 2 for an overview of the evaluation methods used for the SCM and stepped care "lite" studies, including the study population, study design, sample size and outcome measures used. Similar information can be found for the third category of studies in S1 Table. There was great heterogeneity in study design, sample size, and outcomes used. Evaluation methods for each study type are discussed below.

*SCMs:* Only one SCM intervention was evaluated using a design that involved randomization. Specifically, Borsari et al. used a repeated measures design to assess alcohol use outcomes at 3, 6 and 9 months post intervention among those who received step one only (low risk), and those assigned to step two (high risk), who were randomly allocated to receive a BMI or assessment only [51]. By comparing three groups, those at low risk, those at high risk who got the appropriate high-risk intervention, and those at high risk who did not get the appropriate high-risk intervention, not only is the dose-specific response evaluated, but what happens when the hypothesized correct dose is not given is also assessed.

The *Adapted Stepped Care* [12] and *Stepped Care 2.0* [8] models were both evaluated using a system-level pre-post design with focus on service utilization (e.g., number of clients, appointments) [8,12], client satisfaction [8], and step-specific outcomes (e.g., change in scores across psychotherapy sessions) before and after the model was implemented [8,12]. The ES[S]PRIT model was assessed using case study designs focusing on implementation based on utilization data, and average symptom levels for those assigned to each step [50,52]. In the Irish-adapted version [52], utilization patterns of moderate and high-risk participants were compared according to how the program was actually used (e.g., whether moderate-risk participants primarily used the lower intensity interventions, and high-risk participants mostly used the higher intensity interventions). Follow-up outcome data was not included in the evaluation design of either version of the model. Each tier from the STAND intervention was evaluated separately using symptom change scores [54]. Finally, in COMPAS-S an

**Table 2. Summary of Stepped Care Model and Stepped Care "Lite" Studies.**

| Study Type | Study Authors | Intervention Name | Mental Health issues addressed | Student Population | Study Design | N included in analysis | Primary Mental Health Outcome Measures |
|---|---|---|---|---|---|---|---|
| Stepped Care Model | Bauer et al. 2009 [50] | ES[S]PRIT | Eating disorder | College students who have not undergone treatment/have no substantial symptoms for an eating disorder. | Descriptive case study | 357- screening questionnaire; 44 - enrolled into program | -Body Mass Index, SEED AN; SEED BN; WCS; Body dissatisfaction; Overconcern with weight and shape; Unbalanced nutrition and dieting; Binge eating and compensatory behaviours |
| Stepped Care Model | Bailey et al. 2022 [12] | Adapted Stepped Care | Multiple | College students who received counseling through their University Counseling Centre | Analysis of existing data; comparison of pre & post implementation | Full sample = 6313, subsample = 2877 | -Outcome Questionnaire-45 |
| Stepped Care Model | Borsari et al. 2012 [51] | | Alcohol use | Undergraduate students ≥ 18 years who violated campus alcohol policy | Single group repeated measures design with randomization at step 2 | 598 | Alcohol & Drug Use Measure; number of heavy drinking episodes; number of drinks leading up to index event, time spent drinking; YAACQ |
| Stepped Care Model | Cornish et al. 2017 [8] | Stepped Care 2.0 | Multiple | Students using campus-based counselling services | Case studies from three institutions; pre-post data (satisfaction) & comparison across time (counselling outcomes) | 481 for satisfaction data; 785 in counselling treatment group | -Behavioural Health Measure-20/43 |
| Stepped Care Model | Lindenberg et al. 2011 [52] | Appetite for Life (Translation of ES[S] PRIT) | Eating disorder | University students - does not specify otherwise | Implementation evaluation | 78 | WCS, SEED, CR-EAT, EDE-Q |
| Stepped Care Model | Hasking et al. 2023 [53] | COMPAS-S | Suicide risk | All incoming first year students at a large Australian university | Multiple waves of a prospective cohort study | 5454 | -Self-Injurious Thoughts and Behaviors Interview (SITBI), Composite International Diagnostic Interview, 3rd version (CIDI-3.0), Lifetime/past 12 month seeking of mental health treatment |
| Stepped Care Model | Wolitzky-Taylor et al. 2023 [54] | Screening and Treatment for Anxiety and Depression (STAND) | Anxiety; Depression | UCLA students ≥ 18 years, fluent in English, willing to install app on phone | Open trial, quasi-experimental | 4845 screened; 516 received care | -Computerized Adaptive Test-Mental Health (CAT-MH); CAT Depression Inventory (CAT-DI), CAT Anxiety Inventory (CAT-ANX); CAT Suicide Scale (CAT-SS) |
| Stepped Care "Lite" | Bernstein et al. 2017 [20] | | Alcohol use | Mandated students who were first-time violators of alcohol policy | Quasi-experimental, 2 group design | 290 | Drinks per week; Heavy episodic drinking; YAACQ |
| Stepped Care "Lite" | Patrick et al. 2021 [19] | M-Bridge | Alcohol use | First year college students randomly selected | RCT; SMART trial, multiple assignment | 891 | -Frequency of binge drinking; YAACQ |
| Stepped Care "Lite" | Benton et al. 2016 [55] | | Anxiety | Students recruited from campus-based counselling centres with at least moderate BHM-20 anxiety scores. | Quasi-experimental; Comparison to treatment as usual | 1241 | -Behavioural Health Measure-20/43 |
| Stepped Care "Lite" | Lattie et al. 2022 [56] | IntelliCare for College Students | Depression; Anxiety; | Students > 18 years who own a smartphone | Mixed methods; single arm evaluation pre-post implementation | 117 | -PHQ-8; GAD-7 |
| Stepped Care "Lite" | Choi et al. 2023 [41] | *MindYourHead* | Multiple | Chinese-speaking international students | Feasibility study | 130 | -Kessler Distress Scale (K-10); AUDIT-C; Fagerstrom Test for Nicotine Dependence; Perceived Stress Scale |

*(Continued)*

**Table 2.** (Continued)

| Study Type | Study Authors | Intervention Name | Mental Health issues addressed | Student Population | Study Design | N included in analysis | Primary Mental Health Outcome Measures |
|---|---|---|---|---|---|---|---|
| Stepped Care "Lite" | Fitzsimmons-Craft et al. 2021 [16] | Healthy Minds Network | Transdiagnos-tic (Depression; Anxiety; Eating disorder) | Undergraduate students not currently using mental health services and screen positive for clinical anxiety, depression, or eating disorder, or are at high risk for one or more of these. | Protocol for RCT | Expect to enrol 7884 | -PHQ-9; GAD-Questionnaire-IV; Social Phobia Diagnostic Questionnaire, Panic Disorder-Self Report; SWED, EDE-Q; Primary Care Post-Traumatic Stress Disorder Screen, AUDIT-C, self-report drug use; Short Form Survey (SF-12) |
| Stepped Care "Lite" | Jones et al. 2014 [21] | Healthy Body Image Program | Eating disorder | All students in University A; Incoming first-year students in University B. | Implementation and feasibility study | 1551 | -SWED |
| Stepped Care "Lite" | Fitzsimmons-Craft et al. 2019 [57] | Healthy Body Image Program - State Wide Implementation | Eating disorder | Students ≥ 18 years | Observational study | 2454 screenings completed | -SWED |
| Stepped Care "Lite" | Peynenburg et al. 2022 [17] | UniWellbeing Course | Transdiagnos-tic (Depression, Anxiety, Alcohol-use as secondary outcomes) | <18 years old, self-report at least mild symptoms of depression or anxiety on PHQ-9, or GAD-7, | RCT; parallel arm, block randomization with 2X2 factorial design (no control group) | 227 | -PHQ-9; GAD-7; AUDIT-C; Drug Use Disorder Identification Test; Sheehan Disability Scale |
| Stepped Care "Lite" | Currey et al. 2022 [18] | mindLAMP | Depression; Anxiety; | Undergraduate students scoring ≥ 14 on Perceived Stress Scale. | Prospective validation student of a prediction model | 67 | -PHQ-9; GAD-7; Perceived Stress Scale |
| Stepped Care "Lite" | Currie et al. 2010 [58] | Feeling Better | Depression; Anxiety; | Students from psychology subject pool | Development and usability testing | 10 | -Depression Anxiety and Stress Scale (DASS-21) |
| Stepped Care "Lite" | Saekow et al. 2015 [36] | StudentBodies-Eating Disorders (SB-ED) | Eating Disorder | Women | Pilot RCT - two group, online intervention vs. waitlist control | 41 | -EDE-Q, WCS, Clinical Impairment Questionnaire, Center for Epidemiological Studies - Depression Scale (CES-D) |
| Stepped Care "Lite" | Rith-Najarian 2024 [59] | StriveWeekly | Anxiety; Depression | Undergraduate and graduate students ≥ 18 years | RCT - two group, online intervention vs. waitlist control | 1,607 | -Depression Anxiety and Stress Scale (DASS-21) |

Abbreviations in Table 2: Short Evaluation of Eating Disorders - Anorexia Nervosa (SEED-AN); Short Evaluation of Eating Disorders - Bulimia Nervosa (SEED-BN); Weight Concerns Scale (WCS); Young Adult Alcohol Consequences Questionnaire (YAACQ); Clinical and Research Inventory for Eating Disorders (CR-EAT); Eating Disorder Examination Questionnaire (EDE-Q); Alcohol Use Disorders Identification Test - Concise (AUDIT-C); Stanford-Washington University Eating Disorder Screen (SWED); Daily Drinking Questionnaire (DDQ); Patient Health Questionnaire (PHQ-9); Generalized Anxiety Disorder scale (GAD-7); Randomized Control Study (RCT).

intervention and control cohort from the WMH-ICS were compared to determine if those identified as at risk at baseline in the intervention cohort were less likely to remain at risk at the 12-month follow up than the control cohort [53].

Three SCM studies mentioned validating and optimizing decision rules [50,53,78]. The ES[S]PRIT model decision rules for step assignment were validated using symptom data of those invited to participate in higher intensity interventions, and those not [50]. Borsari et al. conducted a follow-up study to assess which individuals responded best to each step, with the goal of understanding how the intervention could be further developed by varying the content and intensity to better meet the needs of individual students [78]. The COMPAS-S algorithm was validated using a different cohort from the WMH-ICS [53].

*Stepped Care "Lite":* Five studies within this category applied randomization [16,17,19,36,59]. Three studies focused on platform usage/engagement as a proxy for intervention dose [16,18,41]. For example, one study determined whether use of modules within the app suggested based on daily mental health assessments was associated with increased engagement and improvement in screening outcomes [18]. Sub-analysis by risk levels according to levels of engagement with the app (e.g., assessments done, links clicked) was also undertaken in the feasibility study of *MindYourHead* [41]; however, outcomes by level of engagement were not assessed. In the protocol for the *Health Mind Network*, it was stated that change in outcomes over time will be assessed by app usage (e.g., number of sessions completed). One study used a system-level pre-post design to assess the number of students using campus-based services based on platform-based recommendations before and after the app *IntelliCare for College Students* was launched, but did not use individual-level data to assess utilization by risk level [56].

One study in this category mentioned validating and optimizing decision rules. Patrick et al. conducted a follow-up study on their staged intervention for alcohol using a Q-learning algorithm to estimate the most optimal decision rules for tailoring variables [79].

*Matching to Mental Health Needs.* Two studies in this category focused on level of engagement with the platform as a proxy for intervention dose [68,72]. Evaluation of an app called *Nod* that provided tailored activities based on mood assessment looked at outcomes by level of engagement [68]. Evaluation of the *BASICS-Mobile app* looked at outcomes by the number of modules received [72]. In another study that provided tailored health information based on drinking profiles, drinker type--which is correlated with the profiles--was compared by study outcomes [60]. In the evaluation of *Space from Depression and Anxiety,* an unguided version of the app that contained no tailoring was compared with a guided version to isolate the effect of tailoring from the core content of the app; however, outcomes associated with individual dose within the guided condition were not assessed [48]. Decomposition was also used by Levin et al., who decomposed the effect of EMA alone compared with EMA with the other tailored intervention components to isolate the effect of symptom tracking [44].

## Student-relevant outcomes

Table 2 provides the mental health outcomes used across the interventions. We were interested in how many studies incorporated outcome measures related to being a student. In total, nine interventions used this type of outcome measure, most often in relation to academics, with specific examples including: academic impairment [37], academic self-efficacy [43], perception of academic functioning [17], GPA [16], enrolment status [16], and intention to stay enrolled [68]. Other examples of student-relevant outcome measures include campus belonging [48,68], fraternity/sorority membership [62], and intention to pledge for "Greek Life."[19].

## Student co-design

Student co-design was most often reported on in the third category of interventions--matching to mental health needs, where nine studies applied it to some extent [43,46–49,62,63,65,68]. Three of the stepped care "lite" interventions mentioned involvement of students in some aspect of co-design [41,56,59]. None of the SCM interventions reported on student co-design. In all cases, it appeared that students were involved in initial design or adaptation work. One intervention indicated that a student "leadership group" was involved [43]. Focus groups were the approach most often used to involve students [47,48,63,65,68].

This rapid scoping review was undertaken to better understand the landscape of SCMs and other related approaches designed for PSS, reflecting the increased interest in SCMs for

addressing what has been framed by some as a "mental health crisis on campus." [4,80,81] Consistent with the review by Berger et al. of SCMs for youth up to 2020, few interventions based on an SCM were found for PSS as a unique population [9]. Specifically, seven interventions were based on a SCM as defined by the intervention designers themselves. However, even within this small group, heterogeneity was observed. Two of the SCMs were integrated system-wide models including an array of mental health services spanning 8 or 9 steps [8,12]. The other five were narrower in scope, made up of a discrete set of interventions across 2 to 4 steps, each addressing particular mental health issues, including depression and anxiety [54], an eating disorder [50,52], problematic drinking [46–49,51,56,62,63,65,68], and suicidal behaviour [53]. The literature is therefore both limited and lacking a consistency in how SCMs are envisioned for PSS.

By applying a wide scope to this review, we were able to identify 13 interventions that, while not identified as SCMs by the intervention designers themselves, could be characterized as stepped care "lite." Almost all interventions identified in this category used a website or app as the first "step", with higher-intensity steps involving direct access or referrals to existing services offered on campus. While spanning fewer steps, these interventions address the requirement of increased access and efficiency, as well as the ability to meet the diverse needs of students. Further, by incorporating symptom tracking, they provide ongoing assessment of mental health needs and the ability to flag those who require higher-intensity care in real time. However, while there is great potential for using technology-driven approaches such as apps, there remains questions as to how students engage with them over the longer term, with studies showing low adherence to app use over time [82,83]. Interviews with PSS have revealed ambivalence toward apps, on the one hand believing in the potential for technology to help but on the other, PSS question the extent to which individuals can effectively manage their mental health on their own [84]. Concerns about privacy and credibility have also been raised [83]. Supported interventions that include therapist check-ins may be more effective, suggesting that integrating technology-driven approaches with traditional counselling services may be the best approach [85]. Therefore, while stepped care "lite" interventions that incorporate technology likely will not replace broader, system-wide models, they may fit into the general landscape of stepped approaches that serve student populations, such as those on waiting lists for services [45], those who could benefit from briefer sessions with a counsellor accompanied by an online platform [55], or those who are not yet accessing mental health services but may require higher-intensity interventions [16].

In a third category of studies, 30 interventions were identified that focused on matching intervention components to the mental health needs of the recipient, but without then changing the intensity of care within the intervention itself. These interventions fit within the landscape of SCMs and stepped approaches, as a key element of SCMs as described by Cornish et al. is the ability to organize delivery of services/resources according to individual mental health needs along with readiness, functioning and capacity to engage [10]. We were interested in how processes for matching to needs might work, and whether evaluation models had been designed to capture outcomes associated with having received an intervention tailored specifically to that individual. Most commonly, these interventions involved providing tailored feedback and information based on symptom profiles developed using data from symptom tracking (e.g., [43]). Several interventions in this group also offered "e-coaching" to increase engagement. Low intensity interventions that provide tailored content fit into the stepped care landscape by offering information and advice that fits the unique needs of the individual. However, striking the right balance between presenting helpful and engaging material and exacerbating the problem with unnecessary embellishments must be considered. "Digital burnout" has been identified as a concern for apps offering notifications [86]. In a

similar vein, research based on the MindLAMP app demonstrated that greater engagement was fostered by having PSS interact with their own data (e.g., symptom tracking) rather than by sending encouraging messages [87]. While it may feel like with ever-newer technology anything is possible, students themselves have indicated that simplicity is important [83], which means avoiding the add-on features and just focusing on the tailored content.

When considering the literature reviewed, we have three recommendations for moving forward research on and implementation of SCMs for PSS. Similar to the call made by Salmon and colleagues for a consistent definition of SCMs for youth [15], we are recommending that a definition of SCMs for PSS needs be developed that is unique to the post-secondary context. This definition must speak to both the unique circumstances of the post-secondary environment, and the needs of students as differentiated from youth more broadly. As for the latter, while PSS and youth are overlapping populations, students experience unique stressors that include those related to achievement, campus culture and financial pressure [88]. Also distinctive from youth populations is the substantial proportion – upward of 30% in some regions -- of PSS that are international students [89]. Providing care to those studying in a country for a limited period of time is complex and requires culturally-adapted approaches [90]. As far as the post secondary environment goes, the definition needs to reflect the fact that campuses are not part of a formal health care system, and generally lack the ability to provide comprehensive assessments, ongoing care, and specialized services such as psychiatry [11]. Therefore, a definition of stepped care developed for youth accessing formal mental health care systems will not fit the circumstances of students accessing care on campus. In addition, with the movement toward whole university approaches that embed considerations of mental health into all aspects of post-secondary life [91], and an increasing focus on salutogenic perspectives on student mental health [92], the definition must be broad enough to consider prevention and population health more broadly. From a student perspective, access to and waiting for care are big concerns [93]. We must therefore consider the margins of care as much as the system of care itself when developing a consensus definition of SCMs for PSS. Future research should consider using a Delphi consensus study, including students, to determine the core components of a definition of SCM for PSS and their evaluation outcomes. Recently, members of our team undertook a Delphi consensus study to understand priorities in student mental health research from a global perspective [94]. Key to the design of the study was including separate groups of professionals, academics and students so that the voice of any one group did not dominate. Results of this current review could be used to develop core components of a definition that could be rated as part of the consensus-building process. While not all interventions identified in this review may be included within a consensus definition, it will be important to consider each with respect to why they do or do not fit.

A second recommendation is that PSIs need not only to undertake evaluations of their locally implemented SCMs, but also draw on their internal research expertise to develop robust designs that provide high quality, publishable data. Without accrual of evidence, it will not be possible to assess the efficacy and provide support for the increasing implementation of this model in PSIs. Following existing recommendations, SCMs should be evaluated as a whole, not at each step separately [15,24]. Another way to think of this is in terms of dose-specific effects, the dose being the specific intervention components (or steps) exposed to. While some attempts were made to assess the overall model using aggregate pre-post data [8,12], or engagement with the website or app as a proxy for intervention dose [16,18,41], the evaluation models reviewed were generally not geared to look at individual-level outcomes capturing the overall effect of the model. This is not an easy undertaking, and will likely require highly pragmatic designs that are co-created with PSS and service providers to integrate practical considerations of the actual setting [95]. Longitudinal cohort studies that

follow students first entering a PSI across multiple years, such as the SWANS cohort study [96], could be used to capture use of an initial intervention, and any subsequent intervention use within an SCM across time. Ideally, PSS would be followed past graduation, to understand longer-term outcomes and the transition from campus-based into adult mental health care systems. Additionally, while randomization is the gold standard for the evaluation of interventions, it is not always clear how to incorporate it in the assessment of stepped care. The two-step SCM described by Borsari et al. was evaluated using a design that involved randomization at step two only. Since all participants received step one, this isolated the effectiveness of stepping up to among those who were determined high risk [51]. Patrick et al. used a sequential multiple assignment randomized trial ("SMART") design involving two randomizations: timing of stage one and, for high-risk participants receiving stage two, the type of intervention provided [19]. One idea for incorporating randomization-like conditions for models with many steps is to make use of a natural experiment. This general approach was undertaken by Hasking et al., who compared two cohorts of students completing the WMH-ICS – one that received an SCM intervention and one that did not [53]. The using of matching techniques could enhance this design to make the cohorts even more comparable. For this work to be feasible, PSIs need to prioritize high-quality evaluation of their own mental health services. In doing so, they should consider how strategic partnerships between academics specializing in disciplines related to student mental health and the professionals involved in the delivery of student services can be developed. At our own institution, a mental health researcher has been embedded into the administrative student mental health team to support high-quality evaluation projects.

The third recommendation is that students must be purposefully and authentically engaged in this work. In our review of the literature, we found that student co-design was often not present, or at least not often reported. When students were involved, the engagement was generally confined to the initial design/adaptation or pre-testing phases rather than being iterative, and little detail was provided to assess the quality of involvement. As student co-design becomes more widely implemented in student mental health research, it will be interesting to assess whether studies that more thoroughly and effectively engage students have better outcomes than those that have minimal student involvement. From a broader perspective, in addition to being involved in studies of SCM interventions, students should be key to helping to define the core components of SCMs for PSS. From a student perspective, this is essential to ensure that when PSIs talk about implementing an SCM, that students know what they are referring to based on a verifiable set of core components. This will also help guide PSIs to implement SCMs that contain the necessary components as endorsed by students themselves. Students also need to be included in the discussion of what outcomes should be focused on for evaluation; this approach had been used by the YouthCan RCT of a stepped care pathway for youth [95]. In our review, we observed that symptom measures were used often, as well as measures like satisfaction and utilization. However, outcomes related to being a student specifically were often missing. We were surprised to find that only a small minority of studies had incorporated functional outcomes such as academic retention and success. While we do not know if students prefer these types of outcomes to the more symptom-focused ones, it is important to find out. Models for youth engagement, such as that used by YouthCan including both high and broad engagement strategies [97] should be implemented in post-secondary settings. A recent review of models for engaging PSS in campus-based mental health research provides specific guidance how this work can be done within a post-secondary context specifically [98]. For example, from a relational perspective, creating a comfortable and safe environment is key, as well as mitigating power imbalances between researchers and the students [98]. From a process perspective, it is recommended that diverse and multiple methods for

collecting feedback are used, small group discussion is facilitated, and clear roles and responsibilities for the students are outlined [98].

This study is not without some limitations. First, as part of our decision to support a rapid approach, we did not review the grey literature. We therefore did not include results from dissertations or research reports not published in the peer-reviewed literature, somewhat limiting the comprehensiveness of the review. Second, since we wanted this review to be broad, we included search terms that would capture features of SCMs over and above the term stepped care specifically. This enabled us to find interventions that were not described as SCMs but contained features consistent with this model. However, it is possible that we did not include enough related terms to rule out the possibly that we missed some interventions that would have fallen within the stepped care "lite" category. Third, the review was limited to English language only, and with evidence of interventions being implemented globally, we likely have missed examples of work being undertaken in some regions. With students moving from region to region to obtain post-secondary qualifications, even PSIs within English-speaking regions need to draw on global data to develop approaches that support students from a variety of cultural backgrounds. Most of the studies reviewed here came from the US, and only one study focused on international students specifically [41], which means that our understanding of SCMs for PSS comes from a limited perspective. It is possible that key features of SCMs may not be equally received by all cultures. For example, the focus on collaboration between care provider and client in SCMs may confront cultural preference for paternalistic care [99]. Future research, including that recommended to develop a consensus definition of SCMs for PSS, should be designed to include a global perspective to ensure this work is applicable to all PSS. Finally, while we put much emphasis on the need to include students in this work, our own inclusion of students was not as extensive as possible. We acknowledge this and are developing a plan to work with students as we move this project forward.

## Conclusions

In conclusion, while SCMs may have great potential to offer benefits to students and institutions [7,8], we need to demonstrate their efficacy before we fully embrace this solution to the exclusion of other evidence-based intervention models. An essential first step to compiling that evidence is agreement on what constitutes an SCM for PSS. This rapid review of the literature demonstrates that the landscape is both limited and lacking consistency in how SCMs are envisioned for PSS. Three recommendations are made to improve research in this area: 1) the need for a consensus definition of SCMs for PSS; 2) the need for more evaluation of local implementation of SCMs using robust methods that can assess dose-specific outcomes; and 3) the need for more extensive involvement of students in co-creating the models and outcomes selected to evaluate them.

## Supporting information

**S1 Table. Summary of matching to mental health needs studies.**
(DOCX)

**S1 Checklist. PRISMA-Checklist.**
(PDF)

**S1 File. PsycInfo Search strategy.**
(DOCX)

**S1 Dataset. Brennenstuhl Stepped Care Scoping Review Dataset.**
(XLSX)

## Author contributions

**Conceptualization:** Sarah Brennenstuhl, Celeste Agard, Kristin Cleverley.

**Data curation:** Sarah Brennenstuhl, Celeste Agard, Rachel Ho.

**Formal analysis:** Sarah Brennenstuhl.

**Investigation:** Sarah Brennenstuhl, Celeste Agard.

**Methodology:** Sarah Brennenstuhl, Kristin Cleverley.

**Project administration:** Sarah Brennenstuhl, Celeste Agard, Rachel Ho.

**Resources:** Kristin Cleverley.

**Software:** Sarah Brennenstuhl.

**Supervision:** Sarah Brennenstuhl, Kristin Cleverley.

**Validation:** Sarah Brennenstuhl.

**Visualization:** Sarah Brennenstuhl, Rachel Ho.

**Writing – original draft:** Sarah Brennenstuhl.

**Writing – review & editing:** Sarah Brennenstuhl, Celeste Agard, Rachel Ho, Kristin Cleverley.

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
