## [Decision Letter · Decision Letter 0]

29 Nov 2024

PONE-D-24-30059Stepped Care, Stepped Care “Lite” & Matching Intervention Components to Individual Mental Health Needs: A Rapid Scoping Review of Mental Health and Substance Use Interventions for Post-Secondary StudentsPLOS ONE

Dear Dr. Brennenstuhl,

Thank you for submitting your manuscript to PLOS ONE. After careful consideration, we feel that it has merit but does not fully meet PLOS ONE’s publication criteria as it currently stands. Therefore, we invite you to submit a revised version of the manuscript that addresses the points raised during the review process.

We look forward to receiving your revised manuscript.

Kind regards,

Hamufare Dumisani Mugauri, Ph.D. Medicine and Health Sciences

Academic Editor

PLOS ONE

3. We note that Figure 2 in your submission contain [map/satellite] images which may be copyrighted. All PLOS content is published under the Creative Commons Attribution License (CC BY 4.0), which means that the manuscript, images, and Supporting Information files will be freely available online, and any third party is permitted to access, download, copy, distribute, and use these materials in any way, even commercially, with proper attribution. For these reasons, we cannot publish previously copyrighted maps or satellite images created using proprietary data, such as Google software (Google Maps, Street View, and Earth). For more information, see our copyright guidelines: http://journals.plos.org/plosone/s/licenses-and-copyright.

1. You may seek permission from the original copyright holder of Figure 2 to publish the content specifically under the CC BY 4.0 license. 

4. As required by our policy on Data Availability, please ensure your manuscript or supplementary information includes the following:

Reviewers' comments:

Reviewer's Responses to Questions

**Comments to the Author**

1. Is the manuscript technically sound, and do the data support the conclusions?

Reviewer #1: Yes

Reviewer #2: Yes

2. Has the statistical analysis been performed appropriately and rigorously? 

Reviewer #1: Yes

Reviewer #2: I Don't Know

3. Have the authors made all data underlying the findings in their manuscript fully available?

Reviewer #1: Yes

Reviewer #2: Yes

4. Is the manuscript presented in an intelligible fashion and written in standard English?

Reviewer #1: Yes

Reviewer #2: Yes

5. Review Comments to the Author

Reviewer #1: 1) The manuscript outlines a rapid scoping review approach, but further clarification regarding the selection criteria for "stepped care lite" interventions would enhance the methodology section. The term is introduced but not thoroughly defined compared to SCM, leading to potential confusion about the inclusion criteria for studies under this category. Expanding on why certain interventions are considered "lite" would improve transparency.

2) The results section notes a wide range of intervention types, including those utilizing apps and symptom tracking. However, the discussion could benefit from a deeper analysis of how technology-driven interventions might impact student engagement or outcomes differently compared to traditional methods. Highlighting specific studies that demonstrate the effectiveness of apps versus in-person interventions would provide a more nuanced perspective.

3) The manuscript identifies several mental health issues addressed by the interventions, including alcohol use and eating disorders. However, there is limited discussion of how these specific mental health issues were prioritized over others like stress or trauma. Providing insight into the selection process and how these disorders were identified as critical areas of focus would add depth to the review’s relevance.

4) The discussion touches on the importance of student co-design, but it would be helpful to elaborate on how student involvement varied across the different interventions. Did interventions that incorporated student input show better outcomes or satisfaction levels? Exploring this relationship could emphasize the practical benefits of co-design and its role in future SCM implementations.

5) While the study briefly discusses limitations, the exclusion of non-English studies is mentioned only in passing. Given the global nature of mental health challenges, the exclusion of non-English studies may limit the applicability of findings in non-English-speaking regions. This limitation could be better addressed, along with a discussion of how cultural factors might affect the generalizability of SCMs.

Reviewer #2: Overall, the study is well designed and prepared.

The methodology demonstrates a well-thought-out and systematic approach. Excellent work in defining clear eligibility criteria that align with the study objectives. The decision to employ a rapid review is well-justified, and the explanation of its suitability for addressing timely mental health needs in postsecondary student populations is commendable.

Involving a multidisciplinary team, including mental health researchers, a clinician-scientist, and an undergraduate student, enriches the study design and enhances the rigor of the definitions and data interpretation.

Excluding interventions that do not align with the study focus (e.g., interventions based solely on preferences) ensures precision and relevance of the findings.

Utilizing Covidence for systematic data management and employing independent reviewers for data extraction and verification ensures reliability and mitigates bias. This process is a strong point in your methodology.

The recommendations are relevant and well-grounded in the findings. However, they could benefit from being more actionable, with specific examples of how PSIs might implement these suggestions.

Emphasizing the lack of student co-design is crucial, but the discussion could expand on strategies or frameworks to meaningfully involve students in SCM development and evaluation. The call for more robust evaluations is essential. Highlighting specific methodologies (e.g., longitudinal studies, SMART designs) strengthens this point, though elaborating on feasibility within PSIs would add value.

The need for a consensus definition of SCMs tailored to PSS is well-justified. Including a brief outline of the proposed Delphi study process could provide clarity and direction. Acknowledging the exclusion of grey literature and non-English sources is commendable. However, a brief mention of how this might affect the review's comprehensiveness would strengthen transparency.

Finally, this study makes a significant contribution to understanding and improving mental health interventions for postsecondary students.

6. PLOS authors have the option to publish the peer review history of their article (what does this mean? ). If published, this will include your full peer review and any attached files.

**Do you want your identity to be public for this peer review?** For information about this choice, including consent withdrawal, please see our Privacy Policy .

Reviewer #1: **Yes: ** Juan Alvarado-Gonzalez MD

Reviewer #2: No

---

## [Author Response · Author response to Decision Letter 1]

24 Jan 2025

Reviewer #1: 

The manuscript outlines a rapid scoping review approach, but further clarification regarding the selection criteria for "stepped care lite" interventions would enhance the methodology section. The term is introduced but not thoroughly defined compared to SCM, leading to potential confusion about the inclusion criteria for studies under this category. Expanding on why certain interventions are considered "lite" would improve transparency.

Thank you for suggesting this. We have revised the definition as follows (see page 8):

A type of multicomponent intervention (see above definition) that provides two or more components based on mental health needs, with at least one component being higher intensity than another, without explicitly stating a SCM was applied. This could include a multicomponent intervention with: a) different models of care (e.g., a self-directed app that steps up to a phone call with a care provider); b) different modalities of the same care model (e.g., group counselling stepping up to individual counselling); or c) different types of information (e.g., information provided to everyone that steps up to an invitation for specific individuals to use services). This definition draws on a core component of Berger’s definitions of SCMs of providing interventions at different levels of intensity[9] and Cornish’s description of SCMs for PSS that includes organizing delivery of services/resources according to the mental health needs of the individual alone or in combination with preferences and level of readiness.[8]

The results section notes a wide range of intervention types, including those utilizing apps and symptom tracking. However, the discussion could benefit from a deeper analysis of how technology-driven interventions might impact student engagement or outcomes differently compared to traditional methods. Highlighting specific studies that demonstrate the effectiveness of apps versus in-person interventions would provide a more nuanced perspective.

Thank you for this suggestion. We have integrated more literature around students’ engagement with apps into several sections of the Discussion as follows (page 21,22):

However, while there is great potential for using technology-driven approaches such as apps, there remains questions as to how students engage with them over the longer term, with studies showing low adherence to app use over time.[82], [83] Interviews with PSS have revealed ambivalence toward apps, on the one hand believing in the potential for technology to help but on the other PSS question the extent to which individuals can effectively manage their mental health on their own.[84] Concerns about privacy and credibility have also been raised.[83] Supported interventions that include therapist check-ins may be more effective, suggesting that integrating technology-driven approaches with traditional counselling services may be the best approach.[85]

Several interventions in this group also offered “e-coaching” to increase engagement. Low intensity interventions that provide tailored content fit into the stepped care landscape by offering information and advice that fits the unique needs of the individual. However, striking the right balance between presenting helpful and engaging material and exacerbating the problem with unnecessary embellishments must be considered. “Digital burnout” has been identified as a concern for apps offering notifications.[86] In a similar vein, research based on the MindLAMP app demonstrated that greater engagement was fostered by having PSS interact with their own data (e.g., symptom tracking) rather than by sending encouraging messages.[87] While it may feel like with ever-newer technology anything is possible, students themselves have indicated that simplicity is important,[83] which means avoiding the add-on features and just focusing on the tailored content.

The manuscript identifies several mental health issues addressed by the interventions, including alcohol use and eating disorders. However, there is limited discussion of how these specific mental health issues were prioritized over others like stress or trauma. Providing insight into the selection process and how these disorders were identified as critical areas of focus would add depth to the review’s relevance.

Thank you for drawing attention to this. Interventions related to trauma would have been included if a primary outcome related to mental health symptoms was used, such as post-traumatic stress, depression, distress etc. To ensure it was clear that our focus was broad, we added post-traumatic stress and self-harm as symptoms in the examples provided in the definition provided. However, the Reviewer is correct, we did not include interventions solely dedicated to stress. This is because stress is generally not considered a mental health problem in and of itself (although it can lead to mental health issues over time). We should note that several interventions included in the review measured stress as an outcome, so it was not completely avoided, just not focused on in isolation from other mental health symptoms. To make the definition of a mental health intervention clearer, we have included the following (page 8):

A stand-alone intervention focusing on improving mental health symptoms (e.g., depression, anxiety, post-traumatic stress, psychological distress, self-harm and suicidal behaviours or symptoms related to an eating disorder, or substance use). Evaluation of the intervention must include at least one mental health or substance use-related measure as a primary outcome. As stress is not generally considered a mental health problem in and of itself, interventions focused solely or primarily on stress to the exclusion other mental health symptoms were not included in the review

The discussion touches on the importance of student co-design, but it would be helpful to elaborate on how student involvement varied across the different interventions. Did interventions that incorporated student input show better outcomes or satisfaction levels? Exploring this relationship could emphasize the practical benefits of co-design and its role in future SCM implementations.

Thank you for this suggestion. Details of student co-design are reported in the Results section on Page 16. Unfortunately, there was not a lot to draw on because the papers reviewed rarely provided sufficient information to assess the quality of student involvement (often student engagement was described in a sentence or less). We have made this limitation more explicit in the Discussion and incorporated your question as to if outcomes vary according to the degree of student involvement as a call for future inquiry as follows (page 24):

When students were involved, the engagement was generally confined to the initial design/adaptation or pre-testing phases rather than being iterative, and little detail was provided to assess the quality of involvement. As student co-design becomes more widely implemented in student mental health research, it will be interesting to assess whether studies that more thoroughly and effectively engage students have better outcomes than those that have minimal student involvement.

While the study briefly discusses limitations, the exclusion of non-English studies is mentioned only in passing. Given the global nature of mental health challenges, the exclusion of non-English studies may limit the applicability of findings in non-English-speaking regions. This limitation could be better addressed, along with a discussion of how cultural factors might affect the generalizability of SCMs.

This is an excellent point. We have elaborated on the limitations regarding drawing on English language studies only and introduced the need to consider cultural factors as follows (page 25):

With students moving from region to region to obtain post-secondary qualifications, even PSIs within English-speaking regions need to draw on global data to develop approaches that support students from a variety of cultural backgrounds. Most of the studies reviewed here came from the US, and only one study focused on international students specifically,[41] which means that our understanding of SCMs for PSS comes from a limited perspective. It is possible that key features of SCMs may not be equally received by all cultures. For example, the focus on collaboration between care provider and client in SCMs may confront cultural preference for paternalistic care.[98] Future research, including that recommended to develop a consensus definition of SCMs for PSS, should be designed to include a global perspective to ensure this work is applicable to all PSS.

Reviewer #2: 

Overall, the study is well designed and prepared. The methodology demonstrates a well-thought-out and systematic approach. Excellent work in defining clear eligibility criteria that align with the study objectives. The decision to employ a rapid review is well-justified, and the explanation of its suitability for addressing timely mental health needs in postsecondary student populations is commendable. Involving a multidisciplinary team, including mental health researchers, a clinician-scientist, and an undergraduate student, enriches the study design and enhances the rigor of the definitions and data interpretation. Excluding interventions that do not align with the study focus (e.g., interventions based solely on preferences) ensures precision and relevance of the findings. Utilizing Covidence for systematic data management and employing independent reviewers for data extraction and verification ensures reliability and mitigates bias. This process is a strong point in your methodology.

Thank you for your kind words.

The recommendations are relevant and well-grounded in the findings. However, they could benefit from being more actionable, with specific examples of how PSIs might implement these suggestions.

Thank you for this great suggestion. We have added some specific examples of how PSIs can make the recommendations actionable. Please see our responses to comment #3, comment #4 and comment #5 for examples of how PSIs can action the recommendations suggested.

Emphasizing the lack of student co-design is crucial, but the discussion could expand on strategies or frameworks to meaningfully involve students in SCM development and evaluation.

This is a good suggestion. We have added some key facilitators for student engagement as follows (page 25):

For example, from a relational perspective, creating a comfortable and safe environment is key, as well as mitigating power imbalances between researchers and the students. [97] From a process perspective, it is recommended that diverse and multiple methods for collecting feedback are used, small group discussion is facilitated, and clear roles and responsibilities for the students are outlined.[97]

The call for more robust evaluations is essential. Highlighting specific methodologies (e.g., longitudinal studies, SMART designs) strengthens this point, though elaborating on feasibility within PSIs would add value.

Thank you for highlighting this point. We have added the following paragraph to the discussion (page 24):

For this work to be feasible, PSIs need to prioritize high-quality evaluation of their own mental health services. In doing so, they should consider how strategic partnerships between academics specializing in disciplines related to student mental health and the professionals involved in the delivery of student services can be developed. At our own institution, a mental health researcher has been embedded into the administrative student mental health team to support high-quality evaluation projects.

The need for a consensus definition of SCMs tailored to PSS is well-justified. Including a brief outline of the proposed Delphi study process could provide clarity and direction.

This is a good suggestion. Co-author Kristin Cleverley has developed a novel design for Delphi studies that stratifies experts into sub-groups to elevate the voice of those traditionally wielding with less influence, such as students. We cite this work and suggest that the review itself be used as the basis for the core components, which are then rated in the consensus process (page 23).

Recently, members of our team undertook a Delphi consensus study to understand priorities in student mental health research.[94] Key to the design of the study was including separate groups of professionals, academics and students so that the voice of any one group did not dominate, and applying a global perspective. Results of this current review could be used to develop core components of a definition that could be rated as part of the consensus-building process.

Acknowledging the exclusion of grey literature and non-English sources is commendable. However, a brief mention of how this might affect the review's comprehensiveness would strengthen transparency.

Another review also brought up this point. Regarding the exclusion of grey literature, we added the following (page 25):

First, as part of our decision to support a rapid approach, we did not review the grey literature. We therefore did not include results from dissertations or research reports not published in the peer-reviewed literature, somewhat limiting the comprehensiveness of the review

Regarding exclusion of non-English sources we have included the following paragraph to better address the limitation (page 25-26):

With students moving from region to region to obtain post-secondary qualifications, even PSIs within English-speaking regions need to draw on global data to develop approaches that support students from a variety of cultural backgrounds. Most of the studies reviewed here came from the US, and only one study focused on international students specifically,[41] which means that our understanding of SCMs for PSS comes from a limited perspective. It is possible that key features of SCMs may not be equally received by all cultures. For example, the focus on collaboration between care provider and client in SCMs may confront cultural preference for paternalistic care.[99] Future research, including that recommended to develop a consensus definition of SCMs for PSS, should be designed to include a global perspective to ensure this work is applicable to all PSS.

Finally, this study makes a significant contribution to understanding and improving mental health interventions for postsecondary students.

Thank you again for your kind words.

---

## [Editor Report · Decision Letter 1]

4 Feb 2025

Stepped Care, Stepped Care “Lite” & Matching Intervention Components to Individual Mental Health Needs: A Rapid Scoping Review of Mental Health and Substance Use Interventions for Post-Secondary Students

PONE-D-24-30059R1

Dear Dr. Sarah,

We’re pleased to inform you that your manuscript has been judged scientifically suitable for publication and will be formally accepted for publication once it meets all outstanding technical requirements.

Kind regards,

Hamufare Dumisani Mugauri, Ph.D. Medicine and Health Sciences

Academic Editor

PLOS ONE
---

## [Editor Report · Acceptance letter]

PONE-D-24-30059R1

PLOS ONE

Dear Dr. Brennenstuhl,

I'm pleased to inform you that your manuscript has been deemed suitable for publication in PLOS ONE. Congratulations! Your manuscript is now being handed over to our production team.

Kind regards,

on behalf of

Mr Hamufare Mugauri

Academic Editor

PLOS ONE